# Biomimetic Hydrogel Applications and Challenges in Bone, Cartilage, and Nerve Repair

**DOI:** 10.3390/pharmaceutics15102405

**Published:** 2023-09-29

**Authors:** Yanbing Gao, Xiaobo Zhang, Haiyu Zhou

**Affiliations:** 1Department of Orthopedics, Lanzhou University Second Hospital, Lanzhou 730030, China; gaoyb21@lzu.edu.cn; 2Key Laboratory of Bone and Joint Disease Research of Gansu Province, Lanzhou 730030, China; 3Department of Orthopedics, Honghui Hospital, Xi’an Jiaotong University, Xi’an 710000, China

**Keywords:** tissue engineering, regenerative medicine, repair and regeneration, biomimetic hydrogel, biomaterials

## Abstract

Tissue engineering and regenerative medicine is a highly sought-after field for researchers aiming to compensate and repair defective tissues. However, the design and development of suitable scaffold materials with bioactivity for application in tissue repair and regeneration has been a great challenge. In recent years, biomimetic hydrogels have shown great possibilities for use in tissue engineering, where they can tune mechanical properties and biological properties through functional chemical modifications. Also, biomimetic hydrogels provide three-dimensional (3D) network spatial structures that can imitate normal tissue microenvironments and integrate cells, scaffolds, and bioactive substances for tissue repair and regeneration. Despite the growing interest in various hydrogels for biomedical use in previous decades, there are still many aspects of biomimetic hydrogels that need to be understood for biomedical and clinical trial applications. This review systematically describes the preparation of biomimetic hydrogels and their characteristics, and it details the use of biomimetic hydrogels in bone, cartilage, and nerve tissue repair. In addition, this review outlines the application of biomimetic hydrogels in bone, cartilage, and neural tissues regarding drug delivery. In particular, the advantages and shortcomings of biomimetic hydrogels in biomaterial tissue engineering are highlighted, and future research directions are proposed.

## 1. Introduction

Tissue defects or injuries caused by external factors such as trauma are one of the most universal problems in surgery [1]. These partial tissue defects or injuries can prevent local blood vessel formation, thus affecting the recovery and rebuilding of the injured part of the body. For example, bone tissue defects caused by peripheral tissue infection, as well as inadequate blood supply, often result in delayed or non-healing bone tissue that is difficult to properly repair [2]. Bone transplantation is the first choice for the treatment of bone tissue defects [3]. However, the lack of bone tissue donors and the immune reactions associated with allografts greatly limit their use [4]. In addition, cartilage tissue has avascular properties, low cell proliferation and migration, and a limited self-repair ability [5]. Therefore, repairing cartilage tissue is also a complex clinical challenge. Meanwhile, neural tissue damage is another difficult problem, usually caused by traumatic brain injury. Particularly in the central nervous system, the regenerative capacity is limited, and tissue regeneration and the recovery of function are generally difficult to achieve after damage [6]. Even with skilled microsurgical intervention, the recovery efficiency is still poor, leading to painful conditions for patients, such as motor and sensory disorders [7]. Given these situations, more and more researchers are looking for a new biomaterial or method to realize the repair of tissue defects or damage based on the structure and construction of personalized modules to overcome existing limitations.

In recent years, tissue engineering, which combines biology and engineering to repair defective tissue or create tissue substitutes, has become a cross-disciplinary field with great development potential. With the development of tissue engineering, hydrogels are increasingly used in biomedicine. Hydrogels have 3D network structural properties and are highly preferred because of their hydrophilic nature, water retention, low toxicity, biocompatibility, biodegradability, and similar structure and properties to natural tissues [8,9,10]. In particular, hydrogels have been used as biological materials to provide support for cell adhesion and facilitate tissue regeneration, as they have exceptional biomimetic properties. Subsequently, diverse approaches have been used to design and synthesize biomimetic hydrogels. So-called biomimetic hydrogels are polymeric materials that mimic the mechanical and biochemical properties of natural tissues and can be autonomously designed and regulated to provide a 3D spatial environment suitable for cell differentiation and survival [11,12]. In general, biomimetic hydrogels are categorized into natural and synthetic polymers based on the source of material properties. Natural biomimetic hydrogels typically include protein-based (e.g., collagen, gelatin, and silk fibroin) and polysaccharide-based hydrogels (e.g., chitosan, alginate, and agarose) [13]. They tend to be characterized by good biocompatibility, low immunogenicity, and easy degradability but have relatively poor mechanical properties and stability [14]. Synthetic biomimetic hydrogels are polymers prepared by the polymerization of artificial monomers, such as polyethylene glycol (PEG), polyvinyl alcohol (PVA), and poly-N-isopropylacrylamide (pNiPAAm) [14]. They not only share the characteristics of natural biomimetic hydrogels but also have excellent mechanical properties and strong stability. Most importantly, the properties of biomimetic hydrogels (e.g., mechanical strength, degradability, mechanical dimensions, swelling properties, and rheology) can be modulated by artificial chemical modifications, combinations of natural and synthetic polymers, the alteration of hydrophilic/hydrophobic ratios or polymer concentrations, as well as by external environmental stimuli (e.g., temperature, time, and light), to achieve a customization of specific biological functions [15,16,17]. It is also worth noting that biomimetic hydrogel drug delivery systems are increasingly favored as an emerging field by a wide range of scholars, and it is precisely through the modulation of different parameters that the functionality can be tuned for therapeutic purposes [18,19]. Therefore, biomimetic hydrogels may be a potential strategy for treating tissue defects or injuries (such as bone, cartilage, and nerve tissue) through their biomimetic properties (Figure 1).

Considering the increasing popularity of biomimetic hydrogels in tissue engineering applications and the gradual increase in the understanding of the properties and functions of biomimetic hydrogels, it is necessary to further review and summarize them. Therefore, we first introduce the preparation methods and characteristics of biomimetic hydrogels in detail. Secondly, we elucidate the application of biomimetic hydrogels in bone, cartilage, and nerve tissue repair and discuss biomimetic hydrogels in separate drug deliveries; for the first time, we provide a detailed comparison between biomimetic hydrogels and biomaterials in bone, cartilage, and nerve tissue applications, and also for the first time, we summarize the biomimetic hydrogels for drug delivery in different areas other than bone, cartilage, and nerve tissues. The advantages and shortcomings of biomimetic hydrogels in biomaterial tissue engineering are then analyzed. Finally, this review provides some directions for the future development of biomimetic hydrogels to provide a theoretical basis for future research.

## 2. The Preparation Method and Characteristics of Biomimetic Hydrogel

Hydrogels can be readily adapted to meet the needs of specific situations based on their physicochemical properties [20]. Hydrogels can be classified into natural and synthetic types depending on the nature of the polymer component of the material [21,22]. However, natural hydrogels have disadvantages such as instability, fast degradation, and low mechanical properties, all of which have restricted their application in tissue engineering [23]. Thus, biomimetic hydrogels can mimic natural tissues or components in tissues to achieve characteristics that resemble the biochemical and physical properties of natural tissues. Biomimetic hydrogels can be synthetic in various ways and can be both physical and chemical hydrogel types through the nature of cross-linking. Physical hydrogels can avoid induced toxicity, while chemical cross-linking can enhance the mechanics and stabilization of hydrogels. However, chemical crosslinkers are limited, since they have poisonous effects on tissues [24]. Radiation crosslinking and enzymatic crosslinking methods avoid the harmful effects of chemical crosslinkers and are highly efficient [25]. Photo-crosslinked hydrogels are probably the most promising, allowing precise spatiotemporal control by adjusting the range, nature, and duration of light exposure, easily changing the mechanical strength of the hydrogel and its biochemical properties [26,27].

The formulation of physical hydrogels is attributed to physical cross-linking interactions, which reduces toxic reactions, as the gelation process usually takes place under mild conditions and without chemical cross-linking agents [28]. Therefore, physical bionic hydrogels designed for the regeneration and repair of tissue defects or injuries are widely preferred. Hydrogels with temperature-triggered properties, on the basis of temperature difference from normal tissue, may respond to thermal stimuli with physical or chemical changes in their shape or internal structure within the body [29]. Especially obvious is the unique swelling–shrinking state transition (i.e., sol–gel state transition) after a temperature change [30]. Poly-N-isopropylacrylamide (PNIPAAm) is a relatively popular material in bone defect repair for its sensitivity to phase changes at temperatures up to 32 °C (lower critical solution temperature, LCST). Through the addition of acrylamide(AAm) to N-isopropylacrylamide (NIPAAm), the LCST is increased from 32 °C to 37 °C, mimicking the human body temperature, to achieve a sol–gel state transition [31]. Alternatively, hydrogels can be likewise prepared via heated/cooled polymer solution [32]. For example, carrageenan behaves as a random coil in hot liquids higher than the melt transformation temperature and as a rigid spiral rod when cooled. Due to the repulsive force of sulfonic acid groups, the double helices additionally combine in the presence of sodium and potassium salts to form stable hydrogels [33]. In addition, ionic cross-linking is formed when a polymer is co-cross-linked by adding divalent or trivalent ions. Minerals, as essential nutrients, are often important for regulating or maintaining cellular behavior, especially copper and zinc [34]. Notably, copper and zinc play a crucial role in maintaining bone mass and promoting the repair of tissue defects or injuries. Recently, a gelatin and copper-zinc bimetallic ion cross-linked hydrogel was produced to mimic the gradient structure of physiological tissues as well as the extracellular matrix (ECM) microenvironment to promote not only osteogenesis and tendon formation but also to facilitate the regeneration of complex tissue structures including fibrocartilage [35]. Bimetallic ion cross-linked hydrogels may serve as a potential biomimetic hydrogel that could offer new strategies for promoting tissue defect or injury repair and regeneration. Finally, there are some hydrogels that have been acquired by hydrogen bonding cross-linking, such as poly(N-vinylpyrrolidone) (PVP)–tannic acid (TA) hydrogels. When PVP and TA solutions are mixed, cohesion between molecules through hydrogen bonding instantly induces gelation, and it is also found that the synthesized PVP–TA cohesive hydrogels exhibit higher adhesion and toughness [36].

Unlike physical crosslinking, chemical crosslinking directly interacts between branched polymers and their polymer chains to form covalent bonds [37]. Among them, chemical cross-linking agents are the most representative—for example, glutaraldehyde and genipin. Glutaraldehyde was used for the preparation of gelatin/alginate hydrogels, while genipin was used for the preparation of chitosan hydrogels [38,39]. Radiation crosslinking synthesizes hydrogels by exposing polymers to direct or indirect radiation via gamma rays and ordinary light beams, producing reactive radicals that interact with polymer chains to form crosslinked networks [37,40]. For example, the synthesis of hyaluronic acid-polyvinyl alcohol hydrogels by gamma radiation facilitates wound healing [41]. In addition, there are many enzymatic cross-linking ways for preparing biomimetic hydrogels for the repair of cartilage tissues. Some scholars have produced injectable bone marrow mesenchymal stem cells (BMSC)-loaded collagen–hyaluronic acid hydrogel crosslinks with enzyme-catalyzed reactions, which improved the physicochemical properties of collagen hydrogels, had high water absorption and retention, and achieved good cartilage tissue repair [42]. Meanwhile, hydrogels can be grafted onto stronger carriers by graft cross-linking to enhance the mechanical performance and structure of ordinary hydrogels with weaker properties. For example, N,N′-methylenebisacrylamide acts as a cross-linking agent to graft acrylic acid onto chitosan (CTS) to obtain chitosan-g-poly(acrylic acid) (CTS-g-PAA) hydrogel, which plays an important role in the horticultural treatment of wastewater dyes [43]. Remarkably, photo-crosslinked hydrogels can be triggered by visible light or UV light [44]. Many polymers of different origins are modified with photo-crosslinking. For example, natural polymers gelatin and synthetic polymers (PEG and PVA) are often used as photo-crosslinked hydrogel materials. Gelatin as a natural polymer is typically modified by photo-crosslinking groups of methacryloyl, acrylamide and norbornene [45]. Gelatin methacrylate (GelMA) is currently a more widely studied photo-crosslinked polymer [46]; GelMA itself has relatively weak osteogenic potential, and the introduction of a bioactive component would give GelMA a desired bone regenerative capacity [47]. Interestingly, Wu et al. recently designed GelMA composite biomimetic hydrogel scaffolds containing calcium carbonate eggshell particle (ESP) reinforcement, and experiments revealed through mechanical testing that the scaffold compression modulus was positively correlated with the ESP concentration; with the increase in ESP, the scaffolds’ mechanical strength increased, which contributed to osteoblasts’ adherence and growth and consequently promoted mineralization. Meanwhile, hydrogels are also water-absorbing, porous polymers with certain swelling properties. Therefore, it was also found that the GelMA hydrogel had a lower water absorption capacity as the amount of ESP increased, resulting in a lower swelling rate. This suggests that this composite biomimetic hydrogel has tunable swelling properties at different ESP amounts [48]. In addition, Huang et al. prepared a temperature-responsive and 3D printable gelatin methacryloyl (GelMA)-poly(N-isopropylacrylamide) (GN) bilayer biomimetic hydrogel, which was successfully crosslinked via photopolymerization of the gel through observational characterization. Surprisingly, the GN hydrogel could undergo temperature-induced phase transition, which was rheologically analyzed to shrink at 37 °C, and recover its initial state at 25 °C, suggesting that the bilayer biomimetic GN hydrogel has tunable mechanical properties and adaptable 3D printability [49]. Wiseman et al. designed a biomimetic self-assembling peptide (SAP) injectable hydrogel—i.e., Fluorenylmethyloxycarbonyl (Fmoc)—conjugated with laminin-1 sequence IKVAV to form a Fmoc-DIKVAV hydrogel, which serves as a scaffold to provide a favorable microenvironment for tissue restoration, anti-inflammation, the support of axon and vascular regeneration, as well as the promotion of astrocyte infiltration, with the potential to serve as a novel modality for the treatment of spinal cord injuries [50] (Table 1).

## 3. The Application of Biomimetic Hydrogel in Bone Tissue Repair

### 3.1. The Application of Biomimetic Hydrogel in Bone Cell Culture

Osteocytes are the most plentiful cells in the skeleton and can act as multifunctional regulators, reshaping the environment around their own lumen and regulating phosphate homeostasis and mechanotransduction [59,60]. Some scholars constructed 3D osteocyte network constructs and performed an in vitro biomimetic mineralization of natural polymers based on bioinks of ECM analogs, i.e., GelMA and hyaluronic acid methacrylate (HAMA) biomimetic hydrogels, which not only enhanced the physical properties of the constructs, but also upregulated osteocyte-related gene expression (e.g., Opg and Sost) [61]. The reconstruction of topologically complex bionic osteoblast hydrogel scaffolds was based on research analysis, which might help to mimic the normal bone tissue microenvironment and make bioprinting clinically relevant. Notably, hydrogels are widely used as 3D cell culture substrates because of their high-water retention and ease of plasticity, which will benefit both mechanism and translation studies. Han [62] et al. synthesized a novel biomimetic hydrogel of monomeric gelatin-norbornene–boric acid (GelNB-BA) by photochemical cross-linking, which not only independently modulates the stiffness and elasticity of the osteoblast matrix but also detects the secreted cytokines associated with encapsulated osteoblasts.

### 3.2. The Biomedical Application of Biomimetic Hydrogel in Bone Tissue Repair

In bone tissue repair, ECM has a critical function in signal transduction and metabolism in tissues [63]. Biomimetic hydrogels that feature 3D space and mimic natural matrices are susceptible to physicochemical modification, can be adjusted further to display suitable mechanical properties, and can bind cells and bioactive substances—making them well-suited scaffolds for cell adherence and differentiation [64,65]. Recently, a loaded mesenchymal stem cell (MSC), nanosilicate (SN), and nanohydroxyapatite (HAP) injectable biomimetic gelatin-methacryloyl hydrogel (GelMA-HAP-SN) system was developed on a rat cranial bone by introducing HAP to confer the physiological tissue similarity of GelMA hydrogel to a normal bone, adding SN to GelMA-HAP hydrogel to confer the desired bone density of the nanocomposite. The SN was added to the GelMA-HAP hydrogel to give the nanocomposites the required bone induction properties and injectability and further modified to achieve biomimetic bone properties. Finally, MSC-encapsulated GelMA-HAP-SN hydrogels were injected into rat cranial defects, and the results confirmed their good bone repair ability and bionic hydrogels as potential grafting materials for the therapy of bone defects [66]. Similarly, Wang et al. designed biomimetic mineralized hydroxyapatite nanofibers (m-HANFs)/methacrylic anhydride-modified gelatin (GelMA) composite hydrogels to mimic the structure and composition of natural bone tissues. Through characterization analysis, the surface of m-HANFs became rougher with increasing days, which was favorable for cell adhesion and inward tissue growth. Meanwhile, the subsequent results showed that the m-HANFs/GelMA hydrogels had a complex porous network structure inside, and the increase in m-HANFs, which was more stable and favorable for nutrient exchange, also improved the mechanical and swelling properties of GelMA and thus promoted the regeneration of bone tissues. The shortcoming of this requires the optimization of the ratio of m-HANFs and GelMA to achieve better bone tissue repair [67]. In addition to hydroxyapatite, tricalcium phosphate (TCP) also plays an indispensable role in bone tissue repair. Wu et al. prepared a biomimetic β-TCP hybrid scaffold immobilized on the surface of a carboxymethyl chitosan (CCS) Hydrogel/Wnt signaling activator (BML)/polydopamine (PDA) interlayer, using dopamine chemistry and the self-assembly method, which was then modified by CCS Hydrogel; the synergistic action of PDA-loaded BML was used to regulate osteogenesis and osteoblastogenesis and was then used to repair bone tissue. Experimental studies with characterization and a 3D structural analysis of this scaffold revealed that the PDA coating caused the color of ceramic TCP to change, and the color did not change after modification with BML and CCS hydrogel coating. Meanwhile, this biomimetic β-TCP hybrid scaffold was shown to have a three-dimensional porous structure and close proximity to each other. Most importantly, this structure was similar to natural cancellous bone. The results suggest that the biomimetic β-TCP hybrid scaffolds provide a three-dimensional porous microenvironment, and the high porosity and porous structure are more conducive to the transport of metabolic wastes, which not only facilitates cell adhesion, growth, proliferation, and promotes osteogenic differentiation but also promotes osteoblastic inactivation and, ultimately, a regeneration of bone tissue [68]. Notably, mesoporous ceramics have emerged as a candidate for bone tissue engineering due to their large surface area as well as controlled drug delivery capabilities. Not only can it be combined with hydrogels allowing for the rapid release of drugs over long periods of time through slow release, but it can also release drugs in response to external environmental stimuli [69]. Studies have shown that mesoporous ceramics combined with biomimetic hydrogel scaffolds that are loaded with drugs or growth factors are important for bone tissue repair [70]. In addition, Datta et al. designed a microsphere-encapsulated biomimetic hydrogel construct that mimics the osteoconductive matrix and dual delivery of osteoinductive factors via microspheres and biomimetic hydrogels for bone tissue regeneration [71]. Such biomimetic hydrogels have the advantage of being slightly traumatic and precisely modulated in the therapy of bone tissue defects. However, the application of these biomimetic hydrogels in bone defects is restricted because of the relatively weak bearing capacity in bone tissue and only for small segments of non-weight-bearing areas. For the improvement of the physical properties of biomimetic hydrogels as scaffolds, pH responsive hydrogels were prepared by Lundberg et al. [72]. The stiffness of the biomimetic hydrogels was greatly improved in the range of pH 5.0 to 8.0 of the biomimetic tissue microenvironment. In addition, strontium (Sr) or nanodiamonds (NDs) were incorporated into the biomaterial hydrogels to biomineralize the bone microstructure [73,74]. This would eventually enhance the physical properties of the biomimetic hydrogel as a scaffold, which has a prospect of application especially in the repair of bone defects in load-bearing areas (Figure 2).

### 3.3. A Comparative Study of Biomimetic Hydrogels and Biomaterials for Bone Tissue Repair

Tissue-engineered scaffolds are temporal inserts that facilitate inward tissue growth and tissue regeneration [75]. They improve tissue defects or healing by appropriately designing geometric, mechanical, and biological properties. Biomaterials are of great importance in tissue engineering scaffolds. Traditional biomaterial bioceramics are produced from diverse materials such as minerals, which are crushed, mixed, molded, and calcined in a series of complex processes [76]. Due to their excellent physical properties and bio-compatible nature as well as their precise chemical composition, they are extensively used for the repair and substitution of bone defective tissues [77]. In recent years, the most popular biodegradable bioceramics are mainly hydroxyapatite (HA) [78] and tricalcium phosphate(TCP) [79], which are not only bioactive and corrosion resistant but can also be degraded by various cell-mediated pathways in bone defects or damaged tissues and eventually replaced by new bone tissue. However, their applications are somewhat limited due to poor toughness, extremely high rigidity, and low strength [80]. Notably, 4D printing technology, however, can be combined with these traditional biomaterials to compensate for certain shortcomings. 4D printing refers to the preparation of 3D objects with physical properties (including shape, density, and electrical conductivity) that can be self-transformable under preset stimuli (e.g., temperature) by using materials that can be altered in programmable ways, such as shape and electrical conductivity, in conjunction with 3D printing technology [81]. A recent study reported a bone morphogenetic protein 2 (BMP-2)-loaded porous nanocomposite scaffold (BMP-2-loaded SMP scaffold) composed of chemically cross-linked poly caprolactone and hydroxyapatite nanoparticles and mimicking a physiological microenvironment. It was shown that the SMP scaffold exhibited good shape-memory effect and cytocompatibility from an extruded shape to a restored shape under 37 °C, which significantly promoted the repair and regeneration of defective tissue in rabbit mandibles [82] (Table 2). In summary, biomimetic hydrogel can combine different biomaterials to improve their physical, chemical, and intrinsic bio-properties, thereby overcoming the shortcomings of traditional biomaterials and improving their biological properties and biocompatibility.

## 4. The Application of Biomimetic Hydrogel in Cartilage Tissue Repair

### 4.1. The Application of Biomimetic Hydrogel in Chondrocyte Culture

Cartilage is arranged on the surface of the bone and allows the joint to move freely by cushioning, absorbing external pressure, or providing a low friction surface [99]. Cartilage is composed of dense ECM and chondrocytes [100]. There are three types of cartilage ECM: hyaline cartilage, fibrocartilage, and elastic cartilage. The percentage of type II collagen in hyaline cartilage is relatively high compared to fibrocartilage [101]. Chondrocytes are resting cells that constitute a very small percentage of cartilage tissue, yet synthesize large amounts of collagen, hyaluronic acid or glycoproteins, and proteoglycans [102]. Hyaluronic acid, in particular, has water retention and is accountable for the high level of cartilage moisture [103]. However, cartilage is an unvascular and unlymphatic tissue, and chondrocytes have a poor proliferative capacity [104]. Therefore, biomimetic hydrogels with simulated structural and biochemical properties are considered to be an effective method for successful chondrocyte regeneration. A hybrid hydrogel of hyaluronic acid and gellan gum was fabricated through chemical and physical cross-linking methods, which is not only highly tough but also has sufficient resistance to extrusion. Moreover, it was shown that this biomimetic hydrogel can promote hyaline-like cartilage regeneration and new ECM deposition as well as support chondrocyte proliferation [105]. In addition, Du et al. mimicked annulus fibrosus (AF) tissue by poly (ε-caprolactone) (PCL) microfibers and alginate hydrogel mimicked and packaged jellied nucleus pulposus (NP) tissue to form mimicked AF-NP composites, which mimicked the structure of natural intervertebral discs (IVDs) and were inoculated with rabbit annulus fibrosus cells (AFC) and nucleus pulposus cells (NPC), respectively. The experimental results revealed that AFC and NPC did not mix with each other, which was also determined by the unique properties of AF and NP tissues. Interestingly, AFC and NPC secreted large amounts of extracellular matrix for better IVD construction, and the compressive modulus of the composites increased over time [106]. Similarly, Liu et al. used GelMA hydrogel to mimic NP tissue and PCL porous scaffolds to mimic AF hierarchical structure, and experiments found that AFC grows well in the scaffolds, and the scaffolds also regulate cell behavior [107]. Another study combined GelMA hydrogel containing chondrocytes with a 3D-printed polylactic acid scaffold to not only mimic the contour, shape, and elasticity of the ear, but also to allow chondrocytes to proliferate [108]. These findings imply that biomimetic hydrogels have excellent mechanical and biological properties that make the regeneration of chondrocytes possible.

### 4.2. The Biomedical Application of Biomimetic Hydrogel in Cartilage Repair

Articular cartilage is a highly hydrated connective tissue that is often destroyed by abnormal external forces. Depending on the collagen fiber arrangement and proteoglycan composition, articular cartilage is classified into four distinct regions [109]. Therefore, when mimicking natural cartilage tissue, each regional layer should be given different mechanical properties as the depth of the joint increases. Recent studies on cartilage tissue biomimetic hydrogels have shown that natural polymers such as gelatin and hyaluronic acid are favored by a wide range of scholars [110]. Hyaluronic acid, in particular, is extensively applied in the manufacture of bionic 3D scaffolds for tissue engineering. A study fabricated composite hyaluronic acid–gelatin phenolized biomimetic hydrogels using a horseradish peroxidase cross-linking reaction, which not only contributed to cell adhesion but also promoted elastic modulus and biological properties [111]. Meanwhile, polyglutamic acid biohydrogel scaffolds with biodegradability and good biocompatibility were designed and prepared by UV initiation, which were experimentally found to enhance both the overall mechanical properties and the expression of cartilage signature genes by BMSC [112]. Additionally, a platelet-rich plasma (PRP)–GelMA 3D biomimetic hydrogel scaffold was designed in a study in rabbits, which mimics the tissue microenvironment and not only participates in dynamic immunomodulation of the M1 to M2 transition of macrophages, but also promotes BMSC osteogenesis and osteochondral differentiation, which facilitates cartilage regeneration and repair [113] (Figure 3). Therefore, this biomimetic hydrogel acts as a scaffold for loading stem cells, does not require other active substances to mediate the differentiation of cartilage tissue, and also induces the polarization of macrophages to M2 and participates in immune regulation. It provides a new strategy for the repair and regeneration of clinical cartilage tissue defects.

### 4.3. A Comparative Study of Biomimetic Hydrogels and Biomaterials for Cartilage Tissue Repair

Compared to bioceramics, metallic materials have good biocompatibility with human cells and tissues. Biocompatibility is the ability of a material to interact with the human body. In the case of metals, this means that they have no adverse effects on the human body and are able to maintain a relatively stable relationship with the human physiological environment. In addition, metallic materials should have mechanical strength that matches that of human tissue. However, metallic materials do not degrade easily, and the rate of degradation is susceptible to the tissue environment [88,89]. Nowadays, with the rapid development of tissue engineering, most traditional biomaterials are modified or designed as scaffolds to incorporate a number of bioactive factors, cells, hormones, and chemicals and to both release and induce targeted cell differentiation and improve cell survival [114]. That is, the three elements used in tissue engineering are the following: scaffolds (e.g., biomimetic hydrogels), seed cells (e.g., mesenchymal stem cells), and bioactive materials (e.g., transforming growth factor beta 1 (TGF-β1), bone morphogenetic protein-2 (BMP-2)) [82,91,114]. Among them, biomaterial scaffolds are mostly pore scaffolds because the porosity and pore size of the scaffold influences the mechanical stability of the scaffold as well as cell adhesion and proliferation, so it has an essential part in tissue repair and regeneration [91]. During the past several decades, numerous traditional methods have been used for scaffolds for tissue repair, like freeze drying, gas forming, electrostatic spinning, solvent casting, and phase separation [115,116]. However, these techniques do not precisely adjust the diverse parameters of the scaffold to the experimenter’s goals and there is a degree of randomization [117]. Therefore, efforts have been made to produce excellent tissue surrogates that can mimic their own tissues to the maximum extent possible, utilizing all possible technologies available. It is worth mentioning that 3D printing technology can be used to create complicated and exclusive 3D structural scaffolds, as it is not only capable of producing porous scaffolds with suitable porosity for tissue defect repair, but also can be architected to have matching of the mechanical properties of the target tissue [118]. 3D printing technologies include solid (fused deposition modeling), powder-based (selective laser melting) and liquid-based (inkjet printing, 3D bioprinting and direct ink writing) techniques [92,93]. A scaffold was fabricated by 3D printing method using poly(N-acryloylglycinamide)/poly(N-[tris(hydroxymethyl)methyl]acrylamide) copolymer hydrogel (PNT) formulated with bioink; the top layer of the scaffold was a hydrogel containing PNT and the growth-active substance TGF-β1; the bottom layer of the scaffold was a hydrogel containing PNT and biomimetic bone-mineralized β-TCP. The results showed that the bionic hydrogel scaffold promoted the regeneration of cartilage tissue and also contributed to the proliferation and differentiation of human bone marrow mesenchymal stem cells (hBMSCs) [91] (Table 2). In contrast, biomimetic hydrogels can combine natural or synthetic polymers and metallic materials to improve the physical, chemical, and intrinsic bio-properties of different biomaterials and to adapt their chemistry and structure to specific needs in order to avoid their disadvantages.

## 5. The Application of Biomimetic Hydrogel in Neural Tissue Repair

### 5.1. The Application of Biomimetic Hydrogel in Neural Cell Culture

The nervous system is mainly classified into the central nervous system (CNS) and the peripheral nervous system (PNS) [119]. Unlike the PNS, the CNS cannot regenerate on its own after injury because it lacks Schwann cells (SCs) that promote axonal growth and forms glial scars that produce an inhibitory microenvironment of ECM components to the detriment of nerve regeneration [120]. SCs are essential for peripheral nerve regeneration; they are specific glial cells that are aligned along axons and are also able to regulate physiological functions and generate appropriate feedback in the release of neurotrophic factors such as nerve growth factor (NGF) and brain-derived neurotrophic factor (BDNF) to support and modulate neuronal function [121]. Autologous nerve grafting is currently the gold standard for nerve repair in clinical practice but is often associated with limited donor availability, size mismatch, secondary donor injury, and tissue adhesions [122,123]. Therefore, it is necessary to find additional approaches for the improvement of nerve regeneration. Currently, biomimetic hydrogels have been applied to neural tissue engineering to mimic the microenvironment of natural neural ECM, and they have many beneficial properties for use in the nervous system, such as tuning the elastic modulus to accommodate the mechanical properties of neural tissue and modification with adhesion ligands to promote cell infiltration and neural axon growth [124,125]. Reduced graphene oxide/gelatin methacrylate anhydride (rGO/GelMA) hybrid biomimetic hydrogels prepared using 3D printing technology and loaded with both SCs and BMSC not only improve the compressive strength and porosity of the hydrogels for cell adhesion but also promote the neuronal differentiation and myelin formation of SCs to achieve the synergistic regeneration of nerve and bone [126]. Thus, biomimetic hydrogels can mimic the natural neural tissue microenvironment and promote the adhesion and differentiation of nerve cells for neural regeneration.

### 5.2. The Biomedical Application of Biomimetic Hydrogel in Neural Tissue Repair

Compared to bone and cartilage tissue repair, nerve tissue repair is more complex and challenging. Research shows that electrical stimulation can facilitate the repair and regeneration of nerve tissue [127]. Therefore, neural tissue engineering has the potential to be an effective method for restoring the functions of the nervous system. Conductive biomaterials have been extensively studied in order to achieve the regeneration of neural tissues. These materials can enhance the cell adhesion and differentiation of neural stem cells to neurons and astrocytes [128,129]. By combining different materials with hydrogels, hybrid conductive biomaterials are capable of being designed for neural tissue repair [130]. Biomimetic hydrogels have also been extensively investigated as cell carriers and drug delivery devices for cell replacement therapies as well as scaffolds to mimic or replace the ECM and support axonal growth. Recently, graphene-sodium alginate (GR-SA) biomimetic hydrogels were prepared to mimic the ECM system with good electrical properties. It was shown that GR-SA hydrogels significantly increased the expression of nerve growth factor (NGF), myelin basic protein (MBP), growth-associated protein-43 (GAP43), and S100 on SCs. MBP and GAP43 have the potential to increase peri-axonal myelin thickness and increase the rate of nerve injury repair, respectively, and are associated with axonal growth and nerve regeneration have important relationships [122]. Therefore, GR-SA biomimetic hydrogels serve as carriers, and mimicking the in vivo microenvironment can promote the adherence and proliferation of nerve growth-associated cells and the ability of periaxonal cell aggregation. In addition, some researchers have prepared aligned fibrin nanofiber hydrogel (AFG) with self-assembling peptides, which can mimic the fibrin cable structure as well as the extracellular matrix of nerve cells and form a microenvironment conducive to tissue regeneration in the existence of neurotrophic factors (NTFs), and they can further fill hollow chitosan tubes to bridge the rat sciatic nerve defect, thus promoting SC cable formation and new axon regeneration and functional recovery [131] (Figure 4). In a study on rats with spinal cord injuries, methacrylate anhydride (MA)-modified HA (HAMA)-collagen-polypyrrole (PPy) nanoparticles (NPs) and a hybrid biomimetic hydrogel modified with MA, loaded with BMSC and conductive antioxidant bifunctional PPy NPs, mimicked the three-dimensional soft mechanical properties and electrical conductivity of the natural spinal cord to provide neuroprotection and a neuroinduction of the 3D ecotone, thereby protecting BMSC from oxidative damage and restoring bioelectrical signaling, inhibiting secondary spinal cord injury, and promoting neuronal differentiation via the synergistic effects of electrical conductivity as well as electrical stimulation, ultimately significantly contributing to neural regeneration and functional recovery [132]. Therefore, biomimetic hydrogels can mimic the central and surrounding nerve tissue microenvironment in a biomechanical and biochemical manner and modulate cellular activity to promote nerve tissue repair.

### 5.3. A Comparative Study of Biomimetic Hydrogels and Biomaterials for Neural Tissue Repair

Undoubtedly, *biomaterials* have been broadly used in the field of tissue engineering. Among them, biodegradable biomaterials have become increasingly popular. Biodegradability means that new tissue can replace filled biomaterials, and the rate of degradation will be in balance with new tissue growth [133]. Thus, biodegradable biomaterials also avoid the harm caused by secondary manipulation and the associated treatment costs. Among them, biodegradable polymers are classified into natural and synthetic polymers [83]. Natural polymers are widely preferred by researchers due to their easy biodegradability, low immunogenicity, and good biocompatibility. For example, collagen is a familiar biomaterial for tissue engineering. Collagen has been used as an internal replacement for nerve conduits to promote sciatic nerve repair in rats [84]. However, they also have the disadvantages of poor stability and low mechanical properties [85]. Therefore, these synthetic polymers can be improved by artificially controlling design and synthesis parameters to enhance polymer kinetic properties [86]. For example, the most common combination is gelatin/polycaprolactone (PCL). Recently, gelatin was successfully hybridized with polylactic acid (PLA) and electrostatically spun to increase differentiation into motor neuron lineages and allow SCs to proliferate in vitro, thereby promoting neurite growth [134,135]. Interestingly, when these synthetic polymers degrade in vivo, they produce acidic decomposition products that change the local tissue microenvironment and thereby adversely affect tissue repair [87]. Additionally, conductive biomaterials on the basis of carbon nanotubes and graphene have been extensively investigated for neural tissue engineering applications due to their high conductivity and flexibility [94]. Among them, carbon nanotubes (CNTs) have strong thermal and electrical conductivity as well as excellent mechanical properties. CNTS are mainly used as implants to promote the growth of nerve protrusions, like the regeneration of the spinal cord after injury in the central nervous system [95]. In turn, CNTS are classified into single-walled carbon nanotubes (SWCNTs) and multi-walled carbon nanotubes (MWCNTs). SWCNTs act as substrates to regulate and stimulate nerve cells through changes in electrical conductivity to restore nerve injury [136]. Conversely, MWCNTs can serve as scaffolds for peripheral nerve repair as well as nerve conduit-targeted drug carriers [96]. In contrast, graphene not only efficiently conducts heat and electricity, but also has a high degree of low cytotoxicity and biocompatibility [137]. 3D graphene substrates have also been reported to be able to encapsulate different gold nanoparticles that increase neuronal differentiation and guide neat axonal alignment [97]. In conclusion, novel biomaterials are a broad group of materials with different chemical structures and physical properties. Their development offers new possibilities for exploring tissue engineering strategies. Particularly when designing tissue scaffolds, it is the selection of suitable degradable novel biomaterials that is the key to tissue repair (Table 2).

## 6. Biomimetic Hydrogels for Drug Delivery in Bone, Cartilage, and Neural Tissues

In recent years, drug delivery mechanisms and tissue engineering have been favored by a wide range of scholars due to their great potential in improving human health diseases. Biomimetic hydrogel is a novel drug delivery method with good biocompatibility and controlled drug release behavior [138]. Biomimetic hydrogels are more promising drug delivery vehicles in the medical field, which can control the rate of drug release according to the different time stages of the disease [139]. In addition, compared to conventional hydrogels, biomimetic hydrogels can sense and respond to small changes in external stimuli—such as temperature, PH, or thermal stimuli—by redesigning the internal network structure and connectivity method of the hydrogel [20]. Therefore, biomimetic hydrogels are also increasingly used in tissue engineering as a substitute material for defect repair or the regeneration of tissues (e.g., bone, cartilage, and neural tissues). Chen et al. designed a PH-responsive carboxymethyl chitosan (CMCh) and amorphous calcium phosphate (ACP) (CMCh-ACP) biomimetic hydrogel, in which the PH was adjusted via gluconolactone δ-lactone acidification to obtain a responsive biomimetic hydrogel. The results showed that the CMCh-ACP biomimetic hydrogel not only supported the proliferative adhesion of loaded MSC but also upregulated the expression of osteogenic markers [140]. In addition, Xin et al. grafted recombinant human bone morphogenetic protein-2 (rhBMP-2) onto mesoporous bioglass nanoparticles (MBGNs) and crosslinked them with GelMA to prepare a composite injectable biomimetic hydrogel, which mimicked the sustained controlled release of proteins from the periosteum, and at the same time, allowed for them to release calcium ions and silica ions after the degradation of the MBGNs, which could promote the proliferation of the cells and synergize with the biologically active factor rhBMP-2 to promote osteogenic differentiation and regulate the repair and regeneration of bone tissue [141]. However, injectable biomimetic hydrogels inevitably produce inhomogeneous injection force during injection, which may affect surrounding healthy tissues. Interestingly, biomimetic hydrogel microspheres prepared by microfluidics have been widely reported for their uniformly dispersed injection force. Recently, Han et al. designed a biomimetic injectable hydrogel microsphere, which was a GelMA hydrogel microsphere surface impregnated with a biomimetic lubrication-coated self-adhesive polymer and encapsulated with diclofenac sodium. The study showed that this biomimetic hydrogel up-regulated the expression of cartilage anabolic genes, which not only improved the lubricity of cartilage tissue but also continuously released anti-inflammatory drugs, which effectively protected the cartilage tissue and exerted the anti-inflammatory effect, thus achieving the goal of alleviating osteoarthritis [142]. Due to the difficulty of replicating the effects of small environmental changes in tissues on biomimetic hydrogel microspheres, this is likely to result in biomimetic hydrogel microspheres being less effective than expected for drug delivery in practice. Yao et al. designed a hyaluronic acid (HA)-pamidronate (Pam)-magnesium (HA-Pam-Mg) injectable composite biomimetic hydrogel, in which controlled release of magnesium ions could activate the PI3K/Akt signaling pathway to promote peripheral nerve function recovery and regeneration. Importantly, the degradation rate of the biomimetic hydrogel has the potential to match the rate of peripheral nerve regeneration [143]. In addition, polylactic acid-glycolic acid (PLGA) is widely known as a drug carrier and has been recognized for its sustained drug release capability. Zarha et al. developed an injectable alginate biomimetic hydrogel based on chelation of metal ions with the addition of a hydrophilic neuroprotective drug, minocycline hydrochloride (MH). The neurogenic regenerative drug paclitaxel (PTX) was then encapsulated in PLGA microspheres and subsequently implanted into the alginate bionic hydrogel to form a dual drug delivery system. The results of the study in rats showed that when the two drugs were delivered over a prolonged period of time, the inflammatory response was reduced, fibrous scarring was reduced, and neuronal repair and regeneration was promoted. This suggests that dual drug delivery successfully improves the microenvironment of neural tissues and has an unexpected therapeutic effect [144] (Figure 5). In conclusion, biomimetic hydrogel drug delivery has been widely used for the repair and regeneration of various tissues. Although these are still in preclinical studies, the use of biomimetic hydrogels as carriers for bioactive factors, drugs, and stem cell delivery for the treatment of various diseases has yielded promising results.

## 7. The Advantages and Shortcomings of Biomimetic Hydrogels in Biomaterials Tissue Engineering

Biomimetic hydrogels have become one of the most important scaffold materials due to their easily modified physicochemical structure, their excellent biocompatibility, and their ability to maintain a three-dimensional network structure. The biomimetic hydrogel as a scaffold offers a suitable growth microenvironment to cells and stimulates the production of natural ECM. For example, stiffer scaffolds alter the cell phenotype and drive MSC differentiation toward osteoblasts [145]. In addition, biomimetic hydrogel can be used for drug delivery to achieve the desired therapeutic effect. Unlike usual oral or intravenous administration, drug-loaded biomimetic hydrogels can be injected directly into the target tissue to release signaling molecules. In a rat model studying spinal cord injury, researchers prepared hyaluronic acid-based hydrogels as three-dimensional biomimetic scaffolds for IKVAV peptides of laminin and brain-derived neurotrophic factors (BDNF) using cross-linking matrix metalloproteinase peptides, injected all grouped samples into the intrathecal space of rats, and monitored them for six weeks. The results showed that the hyaluronic acid-based injectable biomimetic hydrogels containing BDNF created a microenvironment consistent with neural tissue and promoted the differentiation of hMSCs along the neural cell lineage and neural repair and regeneration after a spinal cord injury [146]. Notably, drug-loaded biomimetic hydrogels can control the rate and duration of release not only by modulating the physical and chemical properties of the biomimetic hydrogel but also by releasing the drug biphasically, thus producing therapeutic local drug concentrations. After all, tissue regeneration also needs to occur in a certain correct order. Recently, a nanohydroxyapatite (nanoHA) gelatin biomimetic nanocomposite scaffold coated with electrospun poly(l-lactic acid) (PLLA) reinforced with silica has been fabricated. The expression of the vascular endothelial growth factor (VEGF), which loads angiogenesis, starts at an early stage, whereas BMP-2, which loads and regulates bone and cartilage formation, starts at a later stage. A differential delivery of growth factors is allowed by modulating the hydrophobic structural domains of the composite VEGF pI and BMP-2. That is, VEGF is continuously released in vivo during the first 7 days, while BMP-2 release continues until day 20. The results show that this temporally controlled, differential biphasic release of the biomimetic hydrogel composite scaffold significantly enhances the formation of neovascularization and new bone tissue [147]. In addition to the advantages above, other properties of biomimetic hydrogels include antibacterial properties and immunomodulatory properties. Wu et al. [148] designed polyvinyl alcohol (PVA)/polyvinylidene fluoride (PVDF) composite biomimetic hydrogels to mimic cartilage tissue structure. Meanwhile, Ag-NWs were added to the cartilage layer to induce the formation of β-PVDF phase (piezoelectric phase), which not only improved the piezoelectric properties of PVDF but also imparted certain antibacterial properties to the biomimetic hydrogel. Furthermore, Arlov et al. investigated biomimetic sulfated alginate hydrogels, which inhibited COX-2 protein expression and NF-κB and p38-MAPK activation by sulfation and provided protection against IL-1β induction in human chondrocytes [149]. Thus, immunomodulatory properties provide a protective microenvironment for biomimetic hydrogel-encapsulated cells.

Although biomimetic hydrogels have obvious advantages as scaffolds for tissue engineering, there are still some limitations and shortcomings that need further consideration. First, it needs to be confirmed whether the biomimetic hydrogel has good biocompatibility and whether it is readily degradable in vivo when implanted. Second, biomimetic hydrogels used locally as scaffolds in defective or injured tissues, although with limited systemic toxicity, can also produce inflammatory responses to the host as biomaterials themselves. Third, biomimetic hydrogels regarding nanoparticles may stimulate the immune system and thus produce immune response side effects, and their toxicity needs to be accurately assessed. Fourth, injectable biomimetic hydrogels need to have the properties to form sols and gels and to consider the size, dimensions, and direction of needle entry to avoid a leakage of the needle contents. Fifth, biomimetic hydrogel scaffolds sometimes need to work for long periods of time after insertion, which may adversely affect the surrounding normal tissues. Finally, although some biomimetic hydrogels are based on complex new technologies such as 3D and 4D bioprinting that can precisely regulate the spatial distribution of cells and biomaterials, such technologies may sometimes fail to print more complex geometries to mimic the complex structures of natural tissues, which requires more sophisticated 3D/4D bioprinting technologies and new biomaterials to achieve higher printing resolutions [150,151]. At the same time, most of the experiments are carried out in rodents; when bioprinted biomimetic hydrogel is implanted in these animals, the mechanical behavior of their physiological load is not able to provide human matching geometries to meet the requirements of the clinical applications, which is very detrimental for the translation to a clinical direction [152]. In addition, the use of developed biomimetic hydrogels based on 3D and 4D bioprinting technologies may be limited by the length of the cell cycle, which means that bioprinting complex organs or tissues takes a significant amount of time to complete, which in turn leads to a delay between printing and actual implantation [153]. Notably, the inhomogeneous internal shrinkage of biomimetic hydrogel bioprinting during high-temperature sintering treatments has the potential to damage the scaffolds and also affects drug loading and release under mild conditions [154,155].

Therefore, future research should focus on solving these problems by, for example, improving the biocompatibility and degradability of materials, reducing inflammatory and immune response side effects, optimizing the performance and needle design of injectable biomimetic hydrogels, precisely regulating bioprinting parameters, and exploring more cost-effective preparation technologies. At the same time, the clinical studies of biomimetic hydrogels need to be strengthened to assess their safety and efficacy in order to promote their translation to clinical applications.

## 8. Future Research Directions and Outlook

Biomimetic hydrogels are polymers that mimic biological natural tissues such as ECM and play a vital role in human tissue defects or injuries. In bone tissue, we need to consider the biomaterial source, porous size, and properties of their mechanics to design biomimetic hydrogels for clinical applications that would promote bone tissue repair and regeneration. With regard to cartilage tissue, future research ought to focus on studying the physicochemical and biological properties of biomimetic hydrogels used as scaffolds for tissue regeneration to investigate the effects of stiffness on cell proliferation, differentiation, migration, and ECM production. Additionally, stem cell-based biomimetic hydrogel 3D/4D bioprinting has the potential to open up new avenues for cartilage tissue repair and regeneration. In the neural tissue, the biomimetic hydrogel should be designed and fabricated according to the target biological system and combined with conductive materials to achieve an optimal response. Also, the microscopic arrangement of biomaterials is crucial in neural tissue engineering. Thus, the design of aligned structures within the nerve conduit that can incorporate biomimetic hydrogels offers great promise for neural tissue regeneration. In conclusion, a deeper understanding of the design and preparation of biomimetic hydrogel physicochemical methods is key to the development of materials related to bone, cartilage, and neural tissue engineering. Biomimetic hydrogel 3D/4D bioprinting technology should also be combined with biomaterials or drug delivery systems to meet a range of individualized different stages of therapeutic needs and drug development. In addition, future research should focus on the design of biomaterials that incorporate bionic cues to facilitate repair and regeneration. Studying the underlying mechanisms of biomaterial–tissue interactions in this way can provide inspiration for biomimetic hydrogels in tissue engineering and facilitate the integration of biomaterials with tissues, which will ultimately lead to the design of better biomimetic hydrogels. Therefore, we also have reasons to believe that in the near future, biomimetic hydrogels can be applied in experimental animal models and successfully translated into clinical trials.

## Figures and Tables

**Figure 1 pharmaceutics-15-02405-f001:**
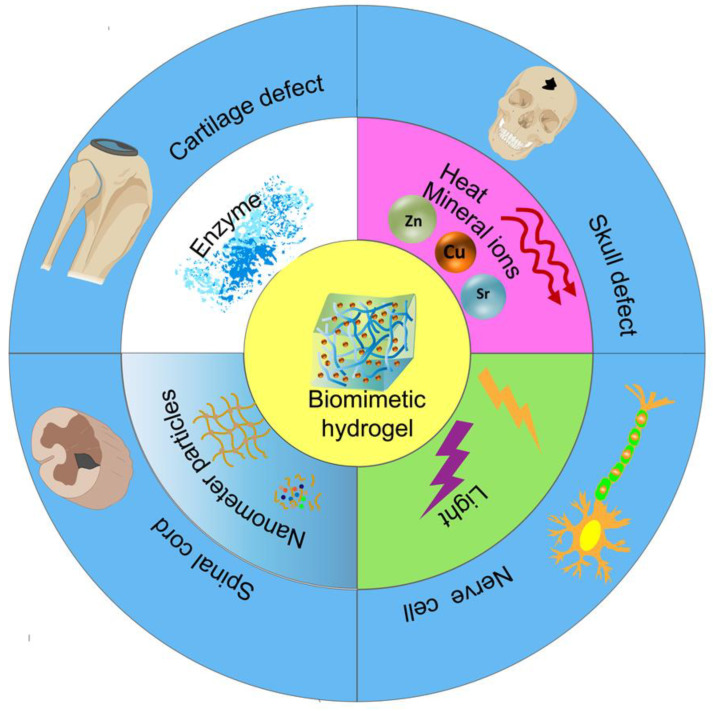
The application of biomimetic hydrogels prepared by different methods in bone, cartilage, and neural tissues.

**Figure 2 pharmaceutics-15-02405-f002:**
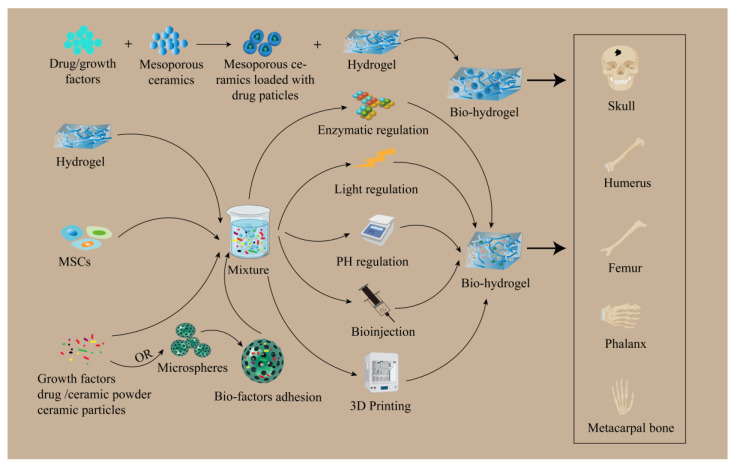
The application of biomimetic hydrogel in bone tissue. Mesoporous ceramic loaded drug particles combined with polymer hydrogel to mimic the structure of bone tissue and slow release of drugs or growth factors to repair bone tissue. Biomimetic hydrogels are used as scaffolds loaded with stem cells, growth factors, drugs, ceramic powders, or ceramic particles and repair bone tissues through enzyme, light, PH regulation, direct injection, or 3D printing. Meanwhile, it can also be embedded with microsphere slow-release bioactive substances in hydrogel scaffolds to repair bone tissues. The research is mainly applied to bone tissues such as the skull, humerus, femur, phalanges, and metacarpal bones.

**Figure 3 pharmaceutics-15-02405-f003:**
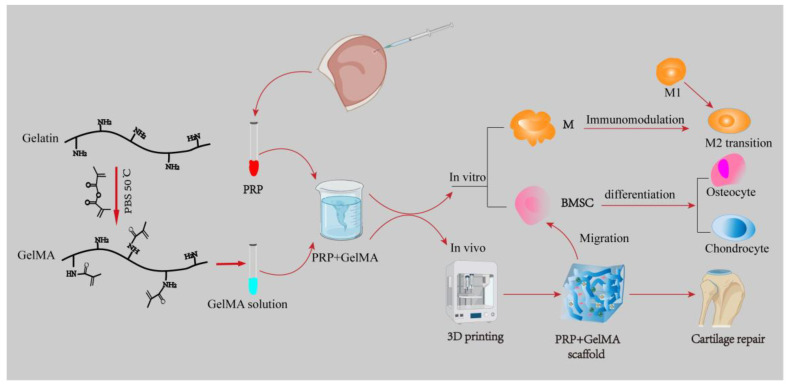
PRP was extracted from whole blood collected in the ear vein of rabbits, gelatin was photo-modified into GelMA, and then the two were mixed to form a PRP–GelMA composite biomimetic hydrogel scaffold. In vivo, it can be used as an implant material by itself to repair cartilage tissue. In vitro, the PRP–GelMA composite biomimetic hydrogel scaffold is involved in macrophage immunomodulation to promote cartilage repair, which not only inhibits macrophage transformation to M1 but also promotes macrophage transformation to M2, and at the same time, it also serves as a bridge to connect with the BMSC, which can migrate to the BMSC to help the cellular differentiation and promotes the formation of osteoblasts and chondrocytes. PRP: platelet-rich plasma; GelMA: gelatin methacrylate; M: macrophage; M1: macrophage1; M2: macrophage2; BMSC: bone marrow stem cell.

**Figure 4 pharmaceutics-15-02405-f004:**
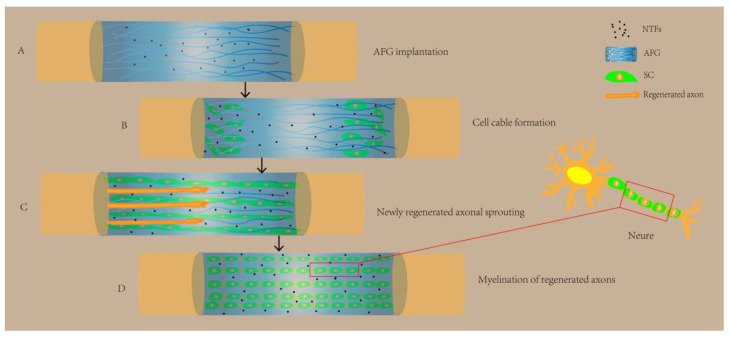
AFG biomimetic hydrogel for neural tissue repair. (**A**) AFG was implanted in hollow chitosan tubes under the condition of NTFs, creating a beneficial environment. (**B**) AFG helped SC cells of nerve stumps to migrate and align in vivo, forming cell cables. (**C**) Guided by the cell cords, axons began to sprout and regenerate. (**D**) Under the combined action of AFG, SC cells, NTFs, and regenerating axons, the myelination were formed, which facilitated the restoration of neurons. NTFs: neurotrophic factors; AFG: aligned fibrin nanofiber hydrogel (AFG); SC: Schwann cell.

**Figure 5 pharmaceutics-15-02405-f005:**
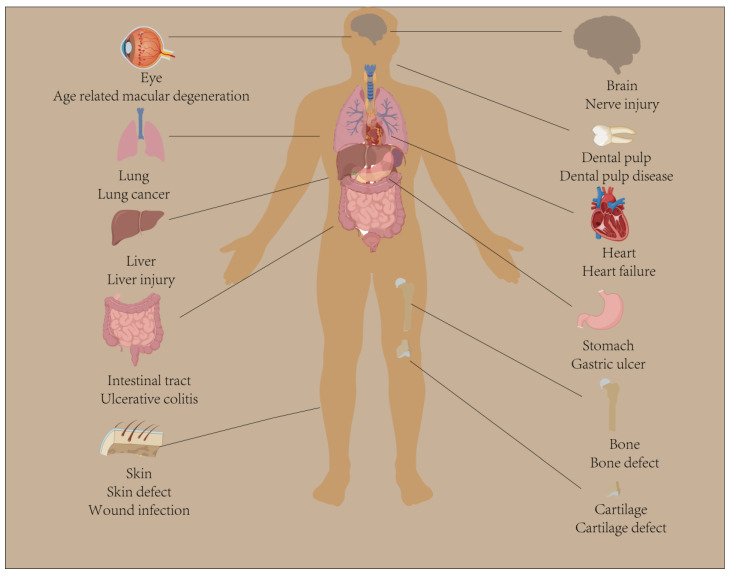
Biomimetic hydrogels involved in drug delivery to organs and tissues. Biomimetic hydrogels as carriers for bioactive factors, stem cells, and drug delivery have been widely studied and applied in biomedical and tissue engineering fields, especially in the repair and treatment of organs or tissues such as bone, cartilage, brain, eyes, lungs, liver, heart, gastrointestinal tract, and skin.

**Table 1 pharmaceutics-15-02405-t001:** Biomimetic hydrogel preparation methods, characteristics, and potential applications.

Biomimetic Hydrogel	Preparation Method	Features	Potential Applications	References
Poly-N-isopropylacrylamide (PNIPAAm)	Temperature stimulus	Temperature influence soluble can make the sol–gel state transform each other.	Tissue engineering.	[31]
Poly(lactic acid)–poly(ethylene glycol) hydrogel	Heating/Cooling	Reversible changes in biochemical properties.	Tissue engineering, drug delivery.	[33,51]
Gelatin polymer gradient Cu-Zn bimetallic ionic hydrogels	Ionic crosslinking	Simulation of physiological tissue and ECM microenvironment.	Tissue engineering.	[35]
Poly-N-vinylpyrrolidone (PVP)–tannic acid (TA) hydrogels	Hydrogen bonding crosslinking	Strong adhesiveness and toughness.	Tissue engineering, antibacterial, hemostatic, wound dressing.	[36,52]
Gelatin/alginate hydrogels, chitosan hydrogels	Chemical agent cross-linking	Improved mechanical properties and stability.	Tissue engineering, drug release.	[38,53,54]
Hyaluronic acid–polyvinyl alcohol hydrogels	Radiation cross-linking	Improves stability and extends the life of hyaluronic acid.	Tissue engineering, wound repair.	[41]
Collagen–hyaluronic acid hydrogel CTS-g-PAA hydrogel Gelatin methacrylate	Enzyme Crosslinking Graft cross-linking Photo-crosslinking	Improving physicochemical properties and water retention. Improved mechanical properties and optimized structure. Accurate manipulation of time and space, facilitating the regulation of biochemical properties.	Tissue engineering, drug development. Horticultural field, controlled release of drugs. Tissue engineering, wound healing, angiogenesis.	[42,55] [43,56,57] [47,58]
ESP/GelMA composite hydrogel	Photo-crosslinking	Promotes osteoblast adhesion, growth, and mineralization; regulates swelling behavior.	Tissue engineering, drug delivery.	[48]
GN hydrogel	Photo-cross-linking/Temperature stimulus	Modulation of mechanical strength and 3D printing adaptation.	Tissue engineering.	[49]
Fmoc-DIKVAV hydrogel	Photo-crosslinking	Anti-inflammatory; supports axonal and vascular regeneration; promotes astrocyte infiltration.	Tissue engineering.	[50]

**Table 2 pharmaceutics-15-02405-t002:** An overview of the strengths and weaknesses of biomimetic hydrogels and biomaterials.

Biomaterials	Strengths	Weaknesses	References
Polymers	Easy biodegradability, low immunogenicity, and good biocompatibility; Biological properties can be improved by designing one’s own synthesis with satisfactory parameters.	Poor stability and low mechanical properties; degradation products have the potential to adversely affect tissue repair.	[83,84,85,86,87]
Bioceramics	Excellent physical properties, biocompatibility, and precise chemical composition; good bioactivity and corrosion resistance.	Poor toughness, very high rigidity, and low strength.	[77,80]
Metallic materials	Good biocompatibility with human cells and tissues and matching mechanical strength; biologically active and conducive to tissue repair and regeneration.	It is not easily degradable, and its decomposition speed is easily affected by the external environment of the tissue.	[88,89,90]
3D printing	A simulation of natural microstructures based on fused deposition modeling, selective laser melting, inkjet printing, and direct ink writing techniques.	High temperatures can damage protein-based biomimetic hydrogels, affecting drug loading and release.	[91,92,93]
4D printing	Excellent shape memory effect and good cytocompatibility.	Slow and inefficient memory response; a lack of material responsive to multiple stimuli.	[81,82]
Carbon nano-tubes/graphene	Mainly used as implants or carriers to promote the restoration of electrical and thermal conductivity of neural tissues.	Toxic; form and dosage may cause adverse reactions in the body.	[94,95,96,97]
Biomimetic hydrogels	It can adjust the chemical structure according to the needs, regulate the decomposition speed and the acidity of the decomposition products, and can be combined with bioactive substances; it can mimic the natural tissue microenvironment to regulate cell behavior.	It can produce immune response side effects; biomimetic hydrogel scaffolds may adversely affect surrounding normal tissues; injectable biomimetic hydrogel needle contents may leak.	[11,12,98]

## Data Availability

The data presented in this study are available in this article.

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
