# Peer review of "Biomimetic Hydrogel Applications and Challenges in Bone, Cartilage, and Nerve Repair"

_pharmaceutics, 2023, doi:10.3390/pharmaceutics15102405_

Round 1
Reviewer 1 Report
This manuscript attempts to review the preparation of biomimetic hydrogels and their characteristics and details the use of biomimetic hydrogels in bone, cartilage, and nerve tissue repair. Since the most useful tissue engineering scaffold materials are now biomimetic hydrogels, this paper may be a welcome addition to the current literature in the field. In my opinion, the idea of this manuscript is good. However, this paper contains some limitations that should be addressed. My concerns and comments are outlined below:
1. Introduction should be more detailed according to types of biomimetic hydrogel-precursor polymers, i.e. synthetics, proteins, polysaccharides, and nucleic acids.
2. There is no claim about what originality is compared to what has been done by others on the same matter. Are there any previous studies that have done similar work?
3. The weakest component of the manuscript is the small number of illustrative and selected figures. Thus, Figures 1 to 4 are considered graphical abstracts.
4. I suggest that the authors identify some relevant figures/graphics and scientific data from recent bibliographic references that would be representative.
5. Table 1 includes only 8 studied biomimetic hydrogels. There are not enough for a review paper.
6. Some properties of such hydrogels were never discussed in the manuscript, such as rheological and swelling properties.
7. In the case of bone repair, do biomimetic hydrogels offer a chemical environment and a surface conducive to new bone formation? This should be highlighted in a recent literature review.
8. Since ceramics are necessary for bone repair, Section 3 should include the use of biomimetic hydrogels with osteoconductive ceramics like calcium phosphate, hydroxyapatite, etc.
9. Figure 2 should be improved and include, in some cases, the use of ceramic powder/particles.
10. Section 4 should include other types of cells in addition to chondrocytes.
11. Figure 3 should be improved and include, in some cases, the use of cells like chondrocytes or others.
12. A new section concerning "the use of biomimetic hydrogels for drug delivery" should be added and highlighted.
13. Section 6 (Comparative study of biomimetic hydrogels and biomaterials) should be moved as a sub-section for each application, i.e. Comparative study of biomimetic hydrogels and biomaterials for bone tissue repair, etc.
14. The authors should be careful with the use of the term "clinical application", and probably "biomedical application" would be more appropriate.
15. The discussions could have been even more useful if any of the commercial products related to biomimetic hydrogels (especially for application in bone, cartilage, nerve repair, and drug delivery) had been highlighted in the manuscript. If the authors can add such information, this manuscript will be useful.
16. Finally, since these hydrogels open doors to an exciting future where 3D/4D bioprinting could revolutionize regenerative medicine, the authors should discuss the limitations on the properties of such hydrogels.
Author Response
Dear reviewer,
Thank you very much for your comments and professional advice. These opinions help to improve academic rigor of our article. Based on your suggestion and request, we have made corrected modifications on the revised manuscript. Please refer to the annex for specific details.

Reviewer 2 Report
This manuscript by Yanbing Gao et al. provides an extensive review of recent developments in the field of biomimetic hydrogels in various medical applications. In my opinion, the manuscript is well written and organised in a logical way. I found no major problems with the technical content, although there were a considerable number of minor issues that should be corrected before publication.
1) Many abbreviations are quoted without adequate explanation where they first appear in the text. Examples include:
AAm, NIPAAm, ECM, PVP, TA, BMSC, GelMA, HAMA, GelMA-HAP-SN, HAP, MSC, PRP, HA(HAMA), PCL, PLA, BMP-2, TGF, TCP, hBMSC,
2) A few sentences were confusing or did not make sense. The authors should check and correct, please:
Bottom of P4 - top of P5: 'For example, N,N'-methylenebisacrylamide act as a cross-linking agent to graft acrylic acid onto chitosan (CTS) to obtain CTS-g-PAA hydrogel, which has an important role in horticulture[36].' This sentence does not accurately reflect what was reported in reference 36. In particular, NN'-methylene-bis-acrylamide does not graft acrylic acid onto chitosan. The authors should check and rephrase, please.
P5: 'Wherein, gelatin, as a photoresponsive polymer...' The authors should check and clarify this statement, please. Gelatin itself is not especially photoresponsive, although it may react with photochemically activated crosslinking agents.
P6: '...confer the desired bone of the nanocomposite...' (might be '...confer the desired bone density of the nanocomposite...')
P7: '...prepared by someonel[56]...' (Who?)
P7: '...in the scope of pH 5.0 to 8.0...' ('...in the range of pH 5.0 to 8.0...' would be better.)
3) Fig. 4: What do A-D mean, please? If it is just intended to indicate progression, adding arrows to the figure may be more intuitive.
Typographical and spelling mistakes:
Fig. 1: 'C' missing from 'cartilage' in upper left quadrant.
P4: 'curl' - 'coil' is the more usual term.
P4: 'dyneolignol' - I do not know - and cannot find out - what this is. It appears that this name is not given in either of the references cited. Is it a spelling mistake? Please check.
P4: '...N,N'-methylenebisacrylamide is act as a cross-linking agent...' ('...N,N'-methylenebisacrylamide acts as a cross-linking agent...' is better.)
P11: 'Ferrum' should be 'iron'.
P11 and table 2: '...immunity...' should be '...immunogenicity...'.
Author Response

(The authors gave the same response as above.)

Reviewer 3 Report
In this review article, Gao et al have reviewed the potential applications of hydrogels in the regeneration of bone, cartilage, and nerve. The topic is exciting and a hot topic in research. In addition, the presentation sounds good. I recommend it for publication. There is only one point, the figures are not of good quality. In many cases, it isn't easy to read the texts in the figures. In addition, the captions of the figures are not illustrative and should be improved.
Author Response
Dear reviewer,
Thank you very much for your comments and professional advice. These perspectives contribute to the academic rigor of our articles. Based on your suggestions and requests, we have made corrections to the revised manuscript. Based on your suggestions, we re-optimized all figures and annotated them in detail.
From all of us at Manuscript, thank you again for taking the time to check out our article!
Reviewer 4 Report
You should review the concepts of biocompatibility of metals in the body, before putting them in hydrogels and recommend their use in Tissue Engineering.
Author Response
Dear reviewer,
Thank you very much for your comments and professional advice. These opinions help to improve academic rigor of our article. Based on your suggestions, we have reworked and added relevant content.Mention exactly where in the revised manuscript this change can be found – page 12, second paragraph, and line8.
From all of us at Manuscript, thank you again for taking the time to review our article!
Round 2
Reviewer 1 Report
All my comments have been addressed properly, and the corrections are acceptable.
However, concerning the limitations on the properties of such hydrogels for 3D/4D bioprinting, the authors should use and add this reference to highlight this point in sections 7 and/or 8:
Natural Hydrogel-Based Bio-Inks for 3D Bioprinting in Tissue Engineering: A Review. Gels 2022 (https://doi.org/10.3390/gels8030179).
Author Response
Dear Editor, Thank you again for taking the time to review our manuscript! Based on your professional comments, we have revised the manuscript as detailed in red in sections VII and VIII of the manuscript.